# Medical Devices with Embedded Sensor Systems: Design and Development Methodology for Start-Ups

**DOI:** 10.3390/s23052578

**Published:** 2023-02-26

**Authors:** Nerea Arandia, Jose Ignacio Garate, Jon Mabe

**Affiliations:** 1Tekniker, Basque Research and Technology Alliance (BRTA), 20600 Eibar, Spain; 2Department of Electronics Technology, University of the Basque Country (UPV/EHU), 48080 Bilbao, Spain

**Keywords:** embedded systems, medical devices, new product development, wearable devices

## Abstract

Embedded systems have become a key technology for the evolution of medical devices. However, the regulatory requirements that must be met make designing and developing these devices challenging. As a result, many start-ups attempting to develop medical devices fail. Therefore, this article presents a methodology to design and develop embedded medical devices while minimising the economic investment during the technical risk stages and encouraging customer feedback. The proposed methodology is based on the execution of three stages: Development Feasibility, Incremental and Iterative Prototyping, and Medical Product Consolidation. All this is completed in compliance with the applicable regulations. The methodology mentioned above is validated through practical use cases in which the development of a wearable device for monitoring vital signs is the most relevant. The presented use cases sustain the proposed methodology, for the devices were successfully CE marked. Moreover, ISO 13485 certification is obtained by following the proposed procedures.

## 1. Introduction

Nowadays, to address the healthcare sector’s challenges, biotechnology is continuously demanding innovation. Recent events such as the COVID-19 pandemic highlight the need to develop more sophisticated, reliable, and connected medical monitoring and diagnostic devices quickly and efficiently. In addition, the ageing of the population and the increase in dependency rates, together with the requirement to monitor and care for patients’ health at their homes, raise the need for new technological systems to deal with these situations.

Embedded connected systems have become a key technology for rapidly developing innovative low-cost diagnostic solutions. Embedded systems can be found in both monitoring and diagnostic systems. Some examples are glucose monitors, pacemakers, wearable monitoring devices, etc.

Being aware of the current opportunity in the sector, an increasing number of biotech start-ups are entering the medical device business. Despite having excellent ideas and technical solutions, many of these start-ups fail. This is due to a lack of knowledge of the healthcare sector and the regulatory requirements that must be met.

The design and development of embedded medical devices are regulated by European Medical Devices Regulations: 2017/745 MDR [1] and 2017/746 IVDR [2]. These two regulations that have entered into force in 2022 to regulate the design and development of these devices with more strict requirements than the previous directives. Devices already on the market must be recertified following these regulations, obliging healthcare companies to adapt their product design and development processes. Noticing the difficulty in adapting to the new requirements, on 16 February 2023, the European Commission decided to extend the deadline, depending on the risk of the device [3]. Other standards define additional aspects.

Many regulatory requirements demand a procedural methodology for performing such developments. After analysing existing product design methodologies, none has been identified yet that fully meets the needs of start-up companies wishing to develop medical devices. None of the identified methodologies is defined in a way that can address the three key characteristics of these developments: compliance with new European medical device regulations, development of innovative products with embedded hardware and software, and inexperienced companies in the sector.

Therefore, as its main contribution, this article presents a methodology that outlines the steps to design and develop embedded medical devices. As it is a methodology for start-ups, the aforementioned methodology aims to minimise financial investment during high-risk technological phases. In addition, it encourages the evaluation of the device by the client or product manager in the early stages of development.

The proposed methodology, intended to be a guide for start-ups in the medical sector, is validated through its application in the development processes of several medical devices. The main use case is the development of a wearable medical device for vital signs monitoring. However, other use cases are presented as references. Many have already obtained the CE marking, thus demonstrating the methodology’s effectiveness. Furthermore, the most critical procedures of the methodology were validated by the certification body Société Générale de Surveillance (SGS) within the ISO 13485 certification process conducted in Tekniker, a Research and Development Centre.

The proposed methodology addresses the needs identified in article [4]. Figure 1 illustrates how the two articles are related.

## 2. Regulatory Context

During medical device design and development phases, compliance with the standards regulating these phases is paramount. Therefore, this section aims to briefly highlight the standards covering the design and development phases of medical devices in Europe. The main regulations are the European Medical Device Regulation (MDR) and the European In Vitro Medical Device Regulation (IVDR), which determine medical devices’ requirements for CE marking. In article [4], the main requirements of each standard are extracted.

The MDR [1] defines the rules and obligations that all manufacturers and distributors must comply with to place a medical device on the European market. On the other hand, the IVDR [2] defines the same aspects for in vitro medical devices. In addition, other standards regulate specific aspects of medical device design and development processes.

The IEC 60601 family of standards [5] defines functional safety and performance requirements for electrical medical devices and equipment. It covers such aspects as electromagnetic compatibility.

The IEC 62304 [6] standard describes the processes associated with all phases of the software lifecycle of a medical device. This standard aims to ensure that medical device software is developed safely.

The ISO 14971 [7] standard defines the requirements for risk management and provides a framework for assessing and controlling the safety risks associated with medical devices. This standard is key for both medical device manufacturers and their suppliers.

Another key standard is IEC 62366-1 [8], which covers the usability of medical devices. This standard helps designers address the device’s usability aspects to ensure that its use involves minimal risk.

ISO 13485 [9] regulates quality management systems for medical devices. To this end, this standard defines the characteristics that a quality management system must fulfil to ensure that these products are reliable and of high quality.

Finally, IEC 81001-5-1 [10] defines the lifecycle and requirements for developing and maintaining medical software to make it cyber secure. The processes defined in this standard are equivalent to those defined in IEC 62304. Therefore, it can be considered that this standard complements IEC 62304 in terms of cybersecurity.

## 3. Review of the Product Design Methodologies

The use of a product design methodology can help to address the regulatory and technical requirements involved in the development of these devices effectively. Therefore, this section reviews existing methodologies for product design. Specifically, it includes Heavyweight methodologies such as Waterfall, V-model, Incremental, Iterative, Spiral and Prototype; Agile methodologies such as Scrum, Kanban and Crystal frameworks; and other methodologies such as Lean Startup, Design Thinking and Stage-Gate. Hybrid methodologies that combine different methodologies are also discussed. For each of the reviewed methodologies, their suitability or difficulty in being implemented for the development of healthcare products is detailed. Finally, after noticing that none of the state-of-the-art methodologies fulfils the nature of the development of embedded medical devices and their regulation, the most relevant requirements or characteristics of each of the current methodologies are extracted. This aims to provide a starting point for the definition of a new methodology for designing and developing embedded medical devices. It is worth noting that when defining these requirements, special consideration is given to the requirements that a methodology must meet to enable a start-up company to successfully develop a medical device.

### 3.1. Heavyweight Methodologies

Heavyweight methodologies are those based on linear and sequential task execution. The basis of these methodologies is to ensure that the development is carried out as planned. They base the success of the projects on deadlines, costs and quality. To accomplish this, development is planned and divided into different phases, which constitute the lifecycle. In general, these methodologies tend to have the following characteristics [11,12]:Predictive approach: Classic management defines in detail the expected output and draws up a development plan, which is used to calculate the cost and deadlines. During execution, monitoring and surveillance activities are carried out to avoid any possible deviations from the plan.Detailed documentation: Developments tend to have a large amount of documentation. This is due to the emphasis on the identification of requirements.Process-oriented: Its objective is to define a process that is able to answer to the needs of the manager, designer, codifier, etc. Each process of the methodology is usually divided into different sub-processes.Universality: Projects, despite their diversity, have common execution patterns. Management practices are based on these characteristics and are valid for any type of project.

#### 3.1.1. Waterfall Methodology

This methodology dates back to 1970 when Winston W. Royce defined the Waterfall model that exists today [13]. The Waterfall methodology is based on a sequential product development process. The project is divided into sequential phases and can only move to the next phase when the previous one has been completed. Its name derives from the sequentiality required by this approach. This methodology includes five phases [14]: requirements definition, design, implementation, verification and maintenance.

Waterfall’s structure is simple and clear [15]. Its use is recommended in cases where the requirements are expected to remain stable throughout development [16]. This allows planning during the initial development phase both in terms of time and cost. In addition, the deliverables generated at each step ensure an efficient flow of information between phases. This, together with its clear and well-defined structure, makes it an easy methodology to control and manage the project. Its clear structure, easy management and the possibility to control deviations make it an ideal methodology for inexperienced teams. On the other hand, being a linear and sequential methodology makes it not very flexible to changes that may arise during the development process [17]. A simple change implies re-executing all the phases, resulting in a very costly process. This methodology focuses on fulfilling established requirements and does not prioritise early customer feedback [18]. For example, in developments involving user interface design, this can become a problem since customer feedback is not received until the late stages of the project. In addition, another difficulty for projects with software components is that testing is completed at a very late stage and only at the system level. Identifying software bugs after the completion of the entire development can lead to a high level of rework [19].

In the case of the development of a new medical device, the verification of critical parts must be completed at the early stages of development. Therefore, this methodology is not suitable for use in developments with high technical uncertainty. The design and development procedures defined in the medical device standards suggest the use of sequential and linear methodologies, such as Waterfall, as they detail a specific order to execute some defined processes [20].

#### 3.1.2. V-Model Methodology

The V-Model methodology is also known as the Verification and Validation Model. Paul Rook defined it in 1980 as an evolution of the Waterfall methodology [21]. As with Waterfall, it is a sequential methodology, requiring the completion of one phase before moving on to the next. In this case, its focus is more on achieving the successful verification of all stages, thus ensuring a high quality of development [22]. This methodology is based on three processes, the first one, the left part of the V-model that defines the conversion of requirements into design, and the second one, the lower part of the V that considers the implementation or execution. Finally, the third process, the right part of the V-model, contemplates the integration and verification of the development [23].

Similarly to Waterfall, it is simple and easy to implement. In contrast to Waterfall, the planning and definition of the verification are completed prior to development, saving time and cost. On the other hand, the execution of verification is envisaged after the implementation of the complete device. This can be inefficient for very complex developments. As with Waterfall, it is a rigid methodology with no flexibility for changes or errors [24]. In addition, this approach does not contemplate using initial prototypes to validate the concept. This methodology is appropriate for adding functionality to an existing product or for the full development of a new product. It is also suitable for use in teams with high technical expertise and in scenarios with low technological uncertainty.

Regarding its applicability to medical device development, the IEC 60601-1 defines as an example a V-model for the development of a Programmable Electrical Medical System (PEMS). Likewise, this model is also adopted by the IEC 62304 standard for the development of medical device software. Therefore, it can be concluded that the most straightforward way to comply with some of the medical device standards is to use the V-model. However, it is not a suitable methodology for start-ups as they do not usually have consolidated technical teams, the product requirements to be developed are not defined from the beginning of the project and the developments, due to their nature, usually present a high degree of technical uncertainty.

#### 3.1.3. Incremental Model

The incremental model seeks a progressive growth of the product’s functionality. That is, the product will grow gradually in each iteration until it reaches the functionality required by the end customer [25]. It is also known as the Successive Version model. The main characteristic of this methodology is that tasks are divided into iterations to achieve the desired functionality [26]. The iterations in this model are not independent; they are linked developments. Unlike in Waterfall and V-model, during the early stages of development, the client can have partial deliverables that allow them to assess how the investment is being materialised. As presented in Figure 2, each development phase is composed of four tasks: requirements, design, implementation, and testing.

In terms of project management, this approach is more complex than Waterfall, as it requires detailed planning, more phases to manage, stronger communication and coordination with the client, and the generation and control of different versions of the development. Likewise, it is necessary to have a clear vision of the final goal of the development, as the initial planning must take into account all the stages of the project. All this makes the cost of development higher than using the Waterfall methodology.

This methodology is an advantageous alternative for medical product development where the overall requirements are clear. It is also suitable when there is a lot of technical uncertainty or the technical team is not highly qualified.

#### 3.1.4. Iterative Model

The Iterative model is based on the iteration of several Waterfall cycles. However, this model does not require having all the specifications of the system to start with the development [27]. The model execution starts with the creation of an initial version of the system. Then, it is evaluated, and further iterations are performed to improve and add functionality to the system. The inputs and outputs of the model are presented in Figure 3.

This customer-oriented model focuses on obtaining user feedback to detect errors in the early stages of development. Effective feedback and model execution require a good communication plan and the client’s involvement as early as possible [28]. It is oriented toward effective risk management as it reduces risk by providing continuous review and redesign of the product [29]. Start-up companies often adopt this methodology to bring their solutions to market quickly, as it allows them to have a functional system solution at the end of each iteration [30].

However, its use for medical products is not advisable, as not considering all the safety requirements at the beginning of the development can lead to a system architecture that does not meet the final requirements, deviating the project from its targeted lead times and cost. The development of medical products involves the generation of a lot of documentation, which further increases as the product evolves, for the performance of certification tests often need to be repeated. This approach can be interesting for developing functionalities that are not an essential part of the medical device and where end-user feedback is important, e.g., for the design of user graphical interfaces.

#### 3.1.5. Spiral Model

The Spiral model was developed by Barry W. Boehm in 1986. It is a combination of the Waterfall model and the Iterative model [31]. It is based on a series of iterations. After the execution of each iteration, a prototype is available on which to assess its risks and suitability. This model focuses on reducing development risks. Typically, it is divided into four phases [32]: determination of objectives and identification of alternative solutions, risk identification and analysis, development and testing, and planning the next iteration.

Completing a task is required to move on to the next step, which makes its execution linear and sequential. At the end of each phase, the required deliverables are generated, easing the transition between phases. In addition, the focus is on receiving customer feedback early in the design process, which requires a continuous dialogue with the customer.

The Iterative and Spiral models have similar applications, but in this case, this model places particular emphasis on risk analysis [33]. As discussed above, the effective development of a medical device relies on accurate risk management. Therefore, this model is more appropriate than the Iterative model for medical device development. Despite this, it presents the same problem as the Iterative model. On the one hand, confused specifications at the beginning of the project can cause deviations in the project deadlines and costs. On the other hand, due to healthcare regulatory requirements, it may not be profitable to launch partial product versions to market.

#### 3.1.6. Prototype Model

The prototype model is based on the development of systems called prototypes. It is an iterative process of trial and error. Starting from an initial specification, a prototype is built and evaluated before an improved prototype is developed [34]. The model, as shown in Figure 4, is composed of six phases [35]: initial requirements, design, prototype building, customer evaluation, prototype refinement and product implementation and maintenance.

This model requires a high degree of customer involvement. Therefore, it is possible to identify whether the development satisfies the user’s expectations in the early stages of development. In addition, errors or deviations can be identified early, and it can help to reduce costs and time compared to the Waterfall model. On the other hand, as with Iterative development, the number of iterations can sometimes increase dramatically, leading to higher costs and delays.

The fact that there are several iterations can result in low-quality documentation. This is a potential problem in medical device development, as documentation is a crucial part of the product certification process. Often, the generation of functional but simplified prototypes can be confusing for the client as they can think that their development is almost ready in advance. This methodology requires good communication with the client so that they understand the current status of the product as well as the remaining phases of the development.

Like the iterative process, it is especially interesting for the development of graphical user interfaces. Through prototyping, the user can experience early in the development stages how the interface will look and make suggestions for changes. It is also effective when the technical solution is not clear; prototyping helps to minimise existing uncertainties.

### 3.2. Agile Methodologies

The origin of Agile methodologies dates back to 1968. This year, a name was given to the cost overruns and quality deficiencies that were being suffered in different software developments [36]. At that time, there was a growing awareness that the requirements were constantly changing in projects. Since then, new methodologies have emerged to work in dynamic and changing environments. Likewise, these methodologies assume that the core of the projects is the people who must be involved in the execution to deal with unexpected events.

Therefore, Agile methodologies were developed to address the huge amount of bureaucracy that traditional methods have to deal with. Agile methodologies are not documentation or development-planning oriented. The fact that they do not dedicate a huge effort to planning makes these methodologies easily adaptable to changes. In addition, these methods are people-oriented and not process-oriented, as they do not focus on developing processes that work for the team; their role is just to support the team [37].

In 2001, several agile design experts created the Agile Alliance, a non-profit organisation to promote agile development and support organisations that wanted to implement these methodologies [38]. As a first step, this alliance drafted the Agile Manifesto. This manifesto is defined to promote better ways of developing software. Although Agile was created for software development, the philosophy promoted by this manifesto can be extended to any other product. To this end, it defines four core values and 12 principles on which this philosophy is based [39].

Agile development methodologies have several concepts that constitute their core. For a better understanding of different agile frameworks, the main terms are briefly detailed [40].

Incremental and Iterative Development: this means that each iteration of product development is functional and adds new functionalities to the product.Backlog: it is a list of developments (new features, changes, corrections, etc.) to be implemented.User Stories: list of product requirements.Daily Meeting: the team holds a daily meeting to identify problems and establish corrective measures.Retrospective Milestone: from time to time, or at the end of the project, all team members spend a day analysing the most significant events that have occurred during the development of the project.

#### 3.2.1. Scrum Framework

Scrum is one of the most extended agile frameworks. It has its origin in a study of new development processes used for successful product design in Japan and the United States [41]. These developments were based on very generic and innovative requirements and had a very short time to market. The Scrum methodology focuses on experience-based learning and self-organisation to address problems and reflection on wins and losses. Although it is considered an agile project management tool, this framework includes meetings and utilities that help teams to structure and manage work. The main elements of Scrum are [42]:Backlog management: Organisation of activities by the product owner is required.Planning: Organisation of the items to be addressed during the iteration (Sprint).Sprint: A period in which the team works together to achieve a development milestone.Daily Scrum meeting: A daily meeting where task tracking is completed. Tasks are monitored through a dashboard where the team has a visual overview of the work. This dashboard presents the list of user stories and the status of different sub-tasks classified as to do, in progress and done.Sprint review: At the end or during the execution of the sprint, the team meets to review the achieved increment.Sprint retrospective: At the end of the sprint, the team meets to assess and share what went right and what went wrong within the iteration.

Scrum has roles and responsibilities. Typically, a team has between five and 15 members, including the Product Owner, the Scrum Master and the development team [43]. The Product Owner is in charge of optimising and maximising the product’s value. Likewise, it is in charge of the Product Backlog and is the link between the team and its stakeholders. The Product Owner must identify and clearly define the objectives in each sprint. The Scrum Master is responsible for ensuring the Scrum is carried out correctly. Therefore, it is responsible for managing the Scrum process and the problems that may arise all along the process. The development team consists of several professionals who are involved in the development of the product. The working team creates the increment for each sprint. In Figure 5, the Scrum workflow is presented.

Agile methodologies, particularly Scrum, constitute a useful tool for start-ups as this methodology offers tools that quickly answer customer needs. It also helps the teams to self-manage to create simplified prototypes. On the other hand, Scrum requires experienced managers to apply the method to be effective. In addition, this methodology is not easy to integrate with the classical project management approach usually employed in medical device development. This characteristic may make this methodology completely suitable for developing medical devices.

#### 3.2.2. Kanban Framework

Kanban is an agile product development framework. It was defined in 1940 by the Toyota engineer Taiichi Ohno. He developed a system for controlling supply chain stocks [44]. This framework is based on the philosophy of continuous improvement, quality assurance, waste reduction and flexibility. It aims to manage development flows in a visual way. To do so, it relies on dashboards [45]. In 2007, Microsoft employee David J. Anderson formulated the four principles and six basic practices of Kanban [46] aligned with the Agile Manifesto.

In terms of roles, two main roles are defined: Service Request Manager and Service Delivery Manager [47]. The Service Request Manager is responsible for identifying customer expectations and needs. It is also responsible for selecting and prioritising the work items. The Service Delivery Manager is responsible for the workflow and must facilitate the Kanban process.

Like Scrum, this framework offers significant advantages for start-ups. It can manage a continuous flow of work input. It supports work input at any time, even with an ongoing task. This approach is more appropriate for team members with work entries that vary in scope and priority. It also supports maintenance and support tasks. On the other hand, Kanban is not a very structured method, so it is harder to implement this framework in a company where employees are not very experienced. Moreover, it does not fit projects that require a high degree of predictability and fully defined planning.

#### 3.2.3. Crystal Framework

Crystal is a framework family for agile development focused on people and their interactions. This methodology was defined in 1991 by Alistair Cockburn, creator of the Agile Manifesto. Specifically, he wrote its first definition for IBM [48]. This methodology pursues individual improvement to achieve the overall improvement of the team. It focuses mainly on people, interaction, community, skills, talent, and communication. The process flow starts from an episode. Episodes are small developments that are the basis for different components. Together, these episodes form an Integration. Iterations are released on a daily or weekly basis. Once the different iterations are finished, the project is integrated, verified, and delivered [49].

It is often used in software development, although it can be generalised to the development of end-to-end products containing software. Depending on the number of people involved in the development, the urgency and the priority of the project, there are different types of crystal methodologies: Crystal clear (less than 8 people), Crystal yellow (10–20 people), Crystal orange (20–50 people) and Crystal red (50–100 people) [48]. Crystal also defines different roles and responsibilities: the Project Sponsor, User Representative, Lead Designer, Programmer, Coordinator, Business Expert, Technical Writers and Testers [50].

This framework provides autonomy and freedom for team members to work in the most efficient way. Crystal reduces management overheads and costs, as there is direct communication between different teams. On the other hand, the lack of processes makes it unsuitable for inexperienced teams, as it can be too complicated and confusing. As the core of the Crystal framework is informal and there is continuous communication between team members, it is not quite compatible with teams that have remote members.

### 3.3. Lean Startup Methodology

Lean Startup is a product design methodology that focuses on customer needs. This method was born as an evolution of the Lean Manufacturing methodology. The origins of Lean go back to 1890, when Sakichi Toyoda, a textile entrepreneur, created several patents for machines that helped automate the work of operators. Kiichiro Toyoda developed this philosophy by focusing on the collaborative work of machines and humans to add value to production without generating waste. This methodology was called Just In Time (JIT).

Lean Startup is a form of Lean defined for new product development by Eric Riese in 2012 in his book *The Lean Startup* [51]. This methodology emerged to create successful companies using continuous improvement. According to Riese, the success of start-ups lies in designing the right processes. Lean Manufacturing is about creating as efficiently as possible, while Lean Startup is based on validating the product concept as quickly as possible. This means creating the product that the customer needs while optimising resources. This method seeks to minimise the risk of failure by ensuring customer feedback at an early stage.

Concept validation is achieved through the development of a Minimum Viable Product (MVP). This concept was defined by Fran Robinson but was not popularised until Eric Ries introduced it in Lean Startup. According to Ries, MVP can be defined as the version of a new product that allows a team to collect the maximum amount of validated learning about customers with the least effort [52]. MVP does not have to be a technological development itself, but it is a way to validate the concept. The Lean Startup methodology is based on three steps that are executed in an iterative manner until the development is successfully completed: Build, Measure and Learn [53].

Agile and Lean Startup methodologies aim to satisfy clients through iterative and incremental processes. The main difference between the two is that while Agile focuses on optimising product development, Lean Startup defines “Measure” and “Learn” as critical parts of its methodology. Lean Startup allows seeing results in the short term, as it optimises the investment of money, reducing risk to a minimum. In addition, it will enable minimum viable products that bring value to the specific market segment. On the other hand, this method needs to consider the administrative and documentation problems involved in developing a medical device.

### 3.4. Design Thinking

Design Thinking is a human-centred approach to product design, innovation and problem solving. The first reference to the Design Thinking methodology dates back to 1969. Herbert Alexander Simon, Nobel Prize winner, mentioned this concept in his book *The Sciences of the Artificial* [54]. It was not until 2008 that Tim Brown defined the methodology as it is known today. This approach was published in an article entitled “Design Thinking” in the *Harvard Business Review* [55].

This methodology is similar to Lean Startup as it includes the development of prototypes or MVPs to define and investigate different solutions. The process followed by this methodology involves the non-linear execution of five steps: empathise, define, ideate, prototype and test. These phases are executed iteratively until the desired development is achieved [56]. In addition, it eases the adoption of the solution, as it is based on the end customer’s needs. It also fosters the team feeling of the employees, as their ideas are valued in the ideate phase. As with Lean Startup, this methodology does not contemplate regulatory and documentary requirements. Therefore, it may be incompatible with the development of medical devices.

### 3.5. State-Gate^®^ Methodology

State-Gate or Phase-Gate is a methodology that assists innovation processes during the generation of new products. It was presented in 1988 in an article published by Robert G. Cooper in *The Journal of Marketing Management* [57]. This methodology is designed to coordinate the creative process through a structure that facilitates investment decisions. To this end, this methodology is divided into six stages or phases and five gates or decision milestones [58]. The stages are the phases of the new product development process and are divided as follows:Stage 0—Discovery: In this stage, ideas and opportunities for development are identified. All stakeholders can participate in this phase. Potential ideas are shortlisted and presented to decide whether they should be taken forward.Stage 1—Scoping: A preliminary scoping of the idea is prepared to evaluate the product and its market.Stage 2—Business plan concept: A detailed analysis of each approach is carried out, assessing its technical, commercial, and economic feasibility. This stage includes the definition and analysis of the product, the creation of the business plan, the project plan and the feasibility review.Stage 3—Development: The prototyping phase of the product and the previously defined plans are executed. A verification and validation plan is also defined.Stage 4—Testing and validation: Product verification and validation are performed. This phase includes all types of testing, such as near testing to identify production errors, field testing or market testing.Stage 5—Launch and implementation: The last phase opens the door to manufacturing and market launch. In this phase, the requirements must be aligned for a successful launch.

Gates are the checkpoints at which a decision is made on whether to continue with the execution of the next phase or stage. These decisions are based on the inputs available at each moment and are made according to different metrics that help evaluate the development’s status. The decisions that can be taken at each gate are Go (go ahead), Kill (do not go ahead), Hold (project in pause), and Recycle (go ahead with changes) [57].

The methodology includes controls related to planning, design and verification and covers the generation of auditable documentation. Both processes are required by the notified bodies responsible for assessing medical devices. In addition, this approach helps to optimise the use of resources, as the entire context is evaluated at each gate. On the other hand, this methodology is not always practical, as the high number of gates can become a bureaucratic obstacle rather than an opportunity for review.

### 3.6. Hybrid Methodologies

In addition to the traditional methodologies, there is a trend to hybridise several of them to achieve more effective and efficient methodologies. The main ones are discussed in the following subsections.

#### 3.6.1. Scrumban Methodology

The Scrumban framework was created from the combination of the benefits of Scrum and Kanban [59]. This methodology emerged when different teams were trying to migrate from one method to the other. During this transition, they identified key aspects that each methodology provided for effective and efficient product development.

On the one hand, Scrum brings the approach of working in well-defined and planned teams to deliver continuous value to end-users. On the other hand, Kanban focuses on process efficiency by ensuring ongoing process improvement. Kanban provides a workflow approach using visual dashboards. Thanks to these features, Scrumban saves time, helps to manage large projects efficiently, helps the whole team to have the same vision, provides fairness among team members and helps to reduce staff stress. On the other hand, this framework is still evolving, and not all the best practices are clearly defined. It also makes project management difficult as long as the whole team is making decisions. It is also unfeasible to monitor the performance of individual team members. Scrumban can be helpful when projects include both product development and maintenance phases. It also works best for teams already using Scrum or Kanban, as they already have many of the principles deeply understood. The Scrumban workflow is presented in Figure 6.

#### 3.6.2. Agile-Stage-Gate

Other hybridisations are based on combining Agile and State-Gate methodologies [60]. This hybridisation offers new features that can make new product development much more efficient. On the one hand, Stage-Gate provides a vision for the selection of ideas or projects to be developed. On the other hand, Agile is more project management-oriented, offering techniques and tools for adaptive, time and cost-optimised development [61].

The mix of the two methods involves Agile working within each State-Gate stage [62,63]. Each stage has a specific duration defined by the typical sprint duration (2–4 weeks). A deliverable is generated at the end of the sprint, and a sprint review is performed through the corresponding gate. This combination is more responsive to customer needs, involves the customer more proactively in the process, reduces iteration time and is much more productive.

Although it is a methodology that can offer many advantages, it is not yet a commonly used method, and there are some gaps or concepts that are still to be defined and evolved. It is challenging to implement in start-up companies without experience in design and development methodologies such as Stage-Gate.

#### 3.6.3. Design-Thinking-Lean Startup-Agile

Another possible hybridisation is the combination of Design Thinking, Lean Startup and Agile [64]. Each of the methodologies is optimal in different processes, so when combined, they can increase the value of the methodology [65].

The process starts with the Design Thinking method, which is suitable for working on the definition of ideas together with the customer. Lean Startup is combined with Agile to validate the hypothesis, define the product business model and develop the customer solution. This solution is generated iteratively through the execution of different sprints. Figure 7 shows a generic process of the methodology. As can be seen, the methodologies are not applied completely, but rather the activities that contribute the most in each phase are selected. Specifically, Design Thinking is used at the beginning of development, and Lean Startup and Agile are combined for the definition and development of the different concepts.

This methodology may be suitable for startups, as it allows for resource efficiency, development according to the specific customer’s needs and early detection of potential problems that may arise. However, it is unclear how to integrate the documentary requirements of medical device regulation into the methodology.

### 3.7. Methodology Contributions

Table 1 lists the main contributions of each of the methodologies to the proposed approach.

## 4. Design and Development Methodology for New Medical Products with Embedded Electronics

Based on the regulatory requirements of embedded medical device development and existing product development methodologies, a new methodology for embedded medical device design and development is presented. In contrast to other existing methodologies, this approach combines elements of project management, new product development models and regulatory aspects. The proposal is based on a methodology that supports the development of new products by optimising the development cost and minimising the associated risk. To this end, a methodology based on three stages is proposed.

The first stage, Development Feasibility, aims to turn new product ideas into technical solutions, thus minimising the technical and economic risk of the solution. The second phase, Incremental and Iterative Prototyping, is based on the Agile development of functional prototypes iteratively and incrementally. In this phase, the main documentation and procedures of the medical device development are established. However, it avoids traditional and rigid medical product development methodologies that do not allow for changes and customer feedback in the early stages. Finally, the third stage, Medical Product Consolidation, proposes the consolidation of the development through a V-model that meets all the regulatory requirements for the design and development of embedded medical devices. This methodology is particularly interesting for start-ups due to the type of solution they usually have, innovative ideas but with high technical and economic uncertainties.

### 4.1. Methodology Audience

This methodology is intended to address the needs faced by medical product design and development teams. It is expected to provide effective uncertainty management and cost control during all design and development phases.

It covers the regulatory requirements for these types of devices and can therefore be applied to any embedded medical device development team. Well-established companies in the sector that already have an in-house design team, research centres that design some or all parts of medical devices for other organisations, or even start-up companies with no experience in healthcare can use this methodology for the design and development of their medical devices.

This methodology focuses on the development of new innovative products. That is, it is envisaged that starting from a preliminary creative idea, it will end up in an embedded medical product. During the initial phases of such projects, there is a high level of uncertainty in both the feasibility of the concept and the specifications, which usually decreases as the technical development progresses. Therefore, this methodology seeks to minimise the investment made during the phases of maximum uncertainty. This methodology is defined in different stages so that by partially executing it, it can also be applied in those developments where both the requirements and the technical solution are clear. The scope of the methodology is presented in Figure 8.

Therefore, it is designed to be used in the design of embedded systems that include some of the following technological blocks: embedded processors, storage units, monitoring or measurement systems, wired or battery power systems, user interfaces and communications interfaces. This is because only the standards regulating these devices have been analysed. In case other applicable standards are identified for a particular design, it will be necessary to review whether this methodology needs to be modified or is directly applicable. It is essential to mention that the reviewed standards are the core standards for any medical device design, so any new standards should be easily integrated.

### 4.2. Methodology Stages

The proposed methodology is based on three phases:Development Feasibility: In the first phase, the methodology aims to create a stage in which different ideas, solutions, and alternatives for developing the concept are identified and validated. The objective of this phase is to minimise the main risks and uncertainties present in the development. To this end, the main concepts are analysed and validated by developing prototypes with minimum functionality.Incremental and Iterative Prototyping: The aim is to develop the first integrated prototype: the first functional solution of the future medical product. The development starts with a prototype with limited functionality, and then different development loops are executed to complete and refine the device’s functionality.Medical Product Consolidation: At this stage, the prototype must evolve into a product. It is important to validate the development to comply with the defined regulation. A significant part of the activity will be devoted to the product’s verification, validation and documentation. For this stage, the methodology proposes to follow a V-model development strategy.

After the execution of each phase, there will be checkpoints or gates to ensure the correct and expected progress of the project. At this point, both the client and the development team will evaluate the progress and suitability of the solution. The checkpoints will also be used to decide whether or not to continue with the proposed development. Figure 9 presents the different phases within the scope diagram of the proposed methodology.

### 4.3. Phase 1—Development Feasibility

In this phase, the idea or concept to be developed is researched and defined by conducting several proofs of concept. As mentioned above, the definition of the intended use and the risk analysis of the device is essential to determine the applicable regulation. Therefore, during this phase, the tasks and approaches required to define the intended use and minimise uncertainties or potential risks must be executed. Typically, these tests are related to the main characteristics of the future medical device. Without the successful implementation of such features, the device could not usually be launched on the market. Figure 10 presents the required inputs for the execution of this phase, the use cases of the development and the generated outputs.

As a starting point, it is common to have a general idea of the development, which is usually related to the measurement principle or sensorisation stage of the future medical device. However, there is rarely a concrete technical idea of how to solve it. In some cases, there is a slight idea that the final solution will be an embedded system with different peripherals such as cameras, antennas, sensor stages, heaters, lighting systems, etc. Usually, solving the measurement or monitoring principle is possible, but its technical feasibility is not guaranteed. Therefore, during this phase, it is intended to define and clarify the parameters to be monitored or controlled for diagnosing or treating a physical condition or disease.

Hence, this phase should include those tests or proofs of concept needed to overcome the main uncertainties of the project—particularly those that could make the future product unfeasible. Therefore, the resulting prototypes will be used for two purposes. On the one hand, they will help to minimise the technical uncertainties faced by the development team. For example, to clarify concerns related to specific parameters of embedded systems, such as performance of the available hardware, reliability of sensing stages, maximum latency, the accuracy of a touch panel, etc.

On the other hand, prototypes can be used to allow the end user to make high-level decisions on the concept. If there is no end-user access, this role is often given to the product manager. For example, suppose a customer wants to introduce a barcode reader. The different typologies will be analysed and tested to select the best to suit the customer’s needs. Suppose a touch panel is to be introduced, but its technology needs to be clarified. The customer must have different minimum viable prototypes to verify capacitive and resistive panels. This way, customer feedback can be obtained to define the best technical solution.

The identified tasks will be listed and specified, each executed by a specialist allowing parallel execution of some of the tasks. However, the feasibility will be validated so that the project manager will foster communication between team members.

#### 4.3.1. Deliverable: Development Feasibility Report

During this phase, the development feasibility document will be generated. The purpose of this document is to create enough information to decide whether the development is feasible. This document should assist in evaluating technical, economic and market feasibility. Although this methodology only contemplates the design and development phases and excludes the business or market that the future device may have, it has to generate information so that the client or product managers can decide on these aspects.

The information included in the development feasibility is summarised below:Use cases: A first approach to the use cases, the main use cases should be identified and written in as much detail as possible.Uncertainties and potential risks: Based on the information provided in the use cases, potential risks and uncertainties to be overcome must be identified at the beginning of the project.Planning and scope: Once the starting point, uncertainties and use cases are known, the scope of the phase must be defined and planned. The acceptance and review of this point by the client are essential, as it helps to ensure a shared development vision.Technical feasibility results: This section will include the tests’ results and the main conclusions on technical feasibility. This information must be clear and precise.Legal feasibility: Information regarding any conflict between the proposed solution and an already registered intellectual property should also be included.Market analysis: The team should search for similar devices already available on the market. The technical solution and its cost will be analysed.Economic analysis: A first approximation of the unit cost of the device and the cost of product development (phase 2 and phase 3) should be made.

#### 4.3.2. Development Feasibility Checkpoint

Once the feasibility phase has been executed, the project manager must deliver the associated deliverables and present the most relevant conclusions to the client. At this point, a decision has to be made on whether to start the development of the first integrated prototype of the future product. This decision usually involves upper management, who typically have absolute control over the activity. In addition, the client company must have other inputs from different departments, such as finance or sales, for final approval.

### 4.4. Phase 2—Incremental and Iterative Prototyping

After clarifying the uncertainties of the project, it is possible to start with the Incremental and Iterative Prototyping phase. This phase aims to design functional prototypes that answer to use cases and requirements defined by the client. As a result, a fully functional and integrated prototype will be available. However, it will be partially verified and documented. The inputs and outputs of this phase are presented in Figure 11.

To this end, the design and development processes use an Agile methodology based on the iterative creation of incremental prototypes. Incremental development is proposed to address the lack of precise requirements and to allow the introduction of requirements and changes in each iteration. In this way, it seeks to reduce the technical and financial risks. An iterative model is proposed to allow the client to assess the suitability of the approach at the end of each iteration. The Agile framework of the proposed model for phase 2 is presented in Figure 12.

This phase is based on the adaptation of the main characteristics of agile methodologies. The key aspects are detailed as follows:Incremental and iterative development: The methodology proposes an incremental and iterative development of the different functionalities of the future product. It starts with those that provide the most outstanding value and are identified, analysed and developed in a parallel way. Once the generated output has been evaluated with the client, iterations are carried out until the desired functionality is completed.Roles: Three main roles will be defined: the product manager or client, the project manager and the development team. The product manager will determine the input requirements for the second phase. It will also define the project backlog and KPIs, the iteration backlog and the retrospective meeting of each iteration. The project manager will accept the requirements proposed by the product manager, manage the project and iteration backlog and define the KPIs. It will also coordinate the iteration follow-up, review and retrospective meetings. The development team will undertake the technical activities and participate in the iteration planning, the follow-up meetings, the definition of the KPIs and the review meeting of each iteration.Key Performance Indicators (KPIs): KPIs will be defined to guarantee the correct progress of the project. They will help objectively measure the development of the project. Two groups of KPIs will be established, one for the project and the other for the iteration. The project KPIs will be defined during a meeting between the project manager and the client or product manager. On the other hand, the iteration KPIs will be parameters that will be defined and measured between the project manager and the team members. The project KPIs will be evaluated in the retrospective meetings between the project manager and the client. On the other hand, the update of the iteration KPIs will take place during the follow-up meetings and their evaluation during the iteration review meetings.Project backlog and user stories: A list of all the activities to be undertaken throughout the project is available. These activities will be carried out in several iterations of the proposed model. Each activity to be executed will be described, prioritised, and categorised according to the functional block to which it belongs. The information about each activity will be known as a user story, and the project manager and the product manager will be responsible for completing and updating it.Iteration planning and iteration backlog: At the beginning of each iteration, the activities to be developed and the deadlines available for them will be defined. The activities executed during an iteration will be identified as the iteration backlog. The product manager and the project manager will decide on the backlog for each iteration. During this meeting, the introduction of possible changes to the established plan will be evaluated. If drastic changes are identified, the whole team will have to analyse the impact and decide whether it is feasible to undertake them. The project manager will be in charge of distributing the tasks among the team members. However, all of the team will be involved during the planning process.Follow-up meeting: Two meetings will be held twice a week to discuss the progress of the tasks. In these meetings, the team will briefly comment on the status of each development and, if any, will explain the problems encountered. These meetings are intended to keep the whole team aware of the progress of the development, update KPIs and take corrective actions to ensure that the development is carried out according to plan. Additionally, these meetings will help to promote communication between the different members of the team. The project manager will be responsible for coordinating this meeting. A digital dashboard will be used as a support tool where the team will update the status of their tasks. The dashboard will present cards with the prioritised and categorised user stories, the status of each task, dates and responsible persons. Figure 13 shows the proposed dashboard format.

Iteration review: After each iteration, a meeting will be held to review the completion of the tasks assigned to that iteration. If some of the activities still need to be completed, these should be re-planned in a subsequent iteration. In addition, during this meeting, the execution of the iteration will be evaluated, and feedback from the whole team will be collected. If necessary, corrective actions will be taken to improve the following iterations and ensure continuous improvement. During this meeting, the progress of the iteration KPIs will also be reviewed. The project manager and the development team will participate in this meeting, and the project manager will be responsible for coordinating it.Iteration Retrospective: After reviewing the progress of the iteration internally (project manager and team members), the conclusions and actions set will be communicated to the product manager or client. The results will then be reviewed with the product manager, and the progress of the project KPIs will be assessed.

#### 4.4.1. Overall Overview of the Proposed Model and Its Deliverables

This section aims to describe the processes of the presented model as well as its execution flow. Although it is represented in a V-shape and not in a circular way, this model contemplates iterations to be able to change or add new requirements, specifications, design or implementation during its execution.

In the case of a change in a user requirement, the entire V-model has to be re-run. On the other hand, if the change affects a software or hardware specification, the specification, design, implementation, and verification phases must be executed, including reviewing the verification plans and their execution. When the change involves a design modification, a review of the development verification plan and the execution of the design, implementation and verification phases shall be performed. Finally, if the implementation changes, the implementation and verification phases must be executed. This methodology, through the execution of partial iteration loops, seeks to facilitate the introduction of changes during development and after the product’s release. The model can be seen in Figure 9.

Although the main objective of this phase is not to ensure compliance with the medical regulations, the proposed model and the generated deliverables are defined to be fully compliant after the execution of phase 3.

The process starts with the definition of the Project Development Plan, which will define the process to be followed during the execution of the project. Then, taking as a starting point the User Requirement Specification (URS), which refers to the definition of the functional and normative requirements, the Project Transference Plan will be defined. This plan contemplates how the project transfer will be carried out, the project acceptance criteria and the definition of the tests that the development team still needs to execute. These tests must be performed by the client, product manager or external team members during phase 3.

Likewise, the transference document must consider when the client will have an output of the development to evaluate it. This methodology proposes generating a set of deliverables for each iteration of the method. These deliverables will serve to assess the progress of the development and clarify technical uncertainties.

Then, knowing the URS and the use cases of the future product, the risks of the product are analysed. The risk analysis and the URS will be the starting point for defining the project specifications: Software Requirement Specification (SRS) and Hardware Requirement Specification (HRS). Once this definition is available, the Software and Hardware tests for the Functional Verification must be defined. These tests will verify each of the defined SRS and HRS. A control milestone will verify that both the definition of specifications and the definition of the associated tests have been carried out correctly.

After the specification phase, the design phase follows. Each HRS and SRS must be translated into a design, a Software Design Specification (SDS) and Hardware Design Specification (HDS). In addition, these design specifications must be verified by Software and Hardware Development Tests, so the test plan must be defined at this point.

Once the design has been made and the associated test defined, a control milestone must be set to verify that progress is as expected. Once the control milestone has been passed, software coding and hardware component manufacturing can proceed.

After the implementation, the defined verification plans must be executed, first the Development Verification Tests, then the Functional Tests, and finally the Transference Tests.

In addition, two transversal processes must also be considered. On the one hand, Configuration Management defines how each project component is formed. On the other hand, the Change Control document is defined, which collects the changes during the design and development phase as well as after the release of the product.

Regarding deliverables, each phase described previously has at least one output that records the activities performed. As the medical device regulation specifies, each step of the design and development processes must be planned and documented; there are a lot of plans and records among the documentation defined within this methodology.

The documentation generated in this phase will be the initial version of the final product documentation. Although the documentation effort is minimised, it should be generated as the development progresses. However, all the information will be consolidated, and the final product documentation will be generated in phase 3.

#### 4.4.2. Incremental and Iterative Prototyping Checkpoint

Phase 2 ends when the V-model is fully iterated with all activities of the product backlog. After this execution, the checkpoint or gate of phase 2 must be executed. At this point, it is evaluated whether the established objectives have been achieved. Additionally, it is decided whether or not to continue with the execution of the development and move on to phase 3 to consolidate the medical product. This decision is usually made by the client or product manager and the company’s executive team. The upper management usually takes into account technical, economic and market aspects when making the decision.

### 4.5. Phase 3—Medical Product Consolidation

The third stage of the methodology aims to consolidate the development and turn it into a medical product. To this end, the deliverables generated during phase 2 are used as input. In this phase, intensive verification is carried out to consolidate and validate both the development and the generated documentation. Inputs and outputs generated in this phase are presented in Figure 14.

Although a fully functional design is available as an input to this phase, it is to be expected that the client or the product manager will want to introduce changes or improvements that have been identified after the delivery of phase 2. Likewise, to validate the development, the design must consider aspects related to its industrialisation. In this phase, the following tasks will be undertaken:Minor modifications or improvements: Even if the main development is completed, minor changes or improvements must be accepted during this phase, such as changes in the colours of the graphical user interface, solutions to minimise risks identified after phase 2 development, etc.Industrialisation-related functionality: During this phase, requirements related to the industrialisation of the product will be added. For this, it will be necessary to involve the production department or, in the case of outsourcing the industrialisation of the product, the Electronic Manufacturing Services (EMS) company that will be in charge of this activity. The industrialisation requirements will be considered an additional use case to the existing ones. Therefore, industrialisation requirements will be added to the URS of the product and will be considered in the SRS/HRS, SDS/HDS, implementation and verification phases. In addition, the project development plan must be updated to indicate how this stage will be undertaken.Intensive verification and validation of the device: An exhaustive verification of the product will be carried out; this verification must cover 100% of the functionality developed by the development team. The validation of the device is also considered during this phase.Product release: The project plan should contemplate how the product release and updates will be performed once the device is on the market.

#### 4.5.1. Overall Overview of the Proposed Model and Its Deliverables

In this phase, it is proposed to follow a V-model similar to the one presented in phase 2. Unlike phase 2, an Agile, Iterative and Incremental model is not considered. Instead, a sequential execution of the V-model is proposed. This will ease product certification by regulatory bodies since the reference standards suggest the development of medical devices following the traditional V-model. Compared to the V-model presented in phase 2, the only difference is that the transference stage becomes a validation stage.

The process starts with the update of the Project Development Plan generated in phase 2. This plan must include how product industrialisation, validation and release will be addressed. Similarly, the roles and scope of processes should be updated where necessary.

Then, the URS must be updated with the use cases to be implemented to undertake the industrialisation phase and the changes requested by the client or product manager. Once the URS has been updated, the validation plan should be generated; this plan will cover the validation of all the defined URS. Tests related to clinical evaluation and research or performance evaluation and performance studies are considered at this stage. Likewise, any testing that applies to the product according to the IEC 60601 family of standards or others must be performed in accredited entities.

The risks of the product are then re-analysed, and, if necessary, appropriate control measures are taken. Once the risk analysis has been performed, the software and hardware specification stages are carried out. This phase must address new functionalities and use cases related to industrialisation. After the specifications are precise, the hardware and software’s functional and environmental verification plan is defined. Based on this, the specification checkpoint must be executed.

Subsequently, the high-level design and detailed hardware (HDS) and software (SDS) design are performed. The development verification plan must also be defined. The development progress is then verified using the design checkpoint. This is followed by implementing the added functionality, both software and hardware. After the implementation, the defined verification plans must be executed, first the development plan, then the functional plan, and finally, the validation plan.

Finally, as in phase 2, configuration management and change control processes must be carried out. Regarding phase 3 deliverables, all of them are presented in Figure 15.

#### 4.5.2. Medical Product Consolidation Checkpoint

The development ends when the proposed V-model is fully executed. Afterwards, the checkpoint or control gate of phase 3 must be completed. At this checkpoint, it will be decided if the design and development have been successful and whether or not to move on to the next phase. This decision is made using information and prototypes generated during the design and development phases, as well as other external elements that help to assess the feasibility of commercialising the product. Usually, market information, feedback from potential users, etc., are used as input. If the assessment is positive, the process of registration and certification of the developed device with the notified bodies must be carried out. This decision is usually taken by the company’s upper management that owns the designed medical device.

## 5. Methodology Validation

After presenting the methodology, this section aims to validate the proposal. On the one hand, the different use cases in which this methodology has been successfully applied are detailed. For each of them, the phases of the methodology that have been applied are specified. However, most of the use cases are associated with industrial clients, so for confidentiality reasons, the description of these use cases is written in a general way without going into technical or client-specific details. Nevertheless, the most relevant information to validate the methodology is presented. Likewise, the use case of a wearable, whose development has been carried out following the proposed methodology, is detailed in more depth. Similarly, it is reported that the methodology has been implemented in Tekniker, where these procedures have been audited during the ISO 13485 certification process.

### 5.1. Use Case 1: Medical Device—Wearable Device Sensor

The wearable solution involves vital signs monitoring platform in the form of a smart wristband, which can collect relevant data about the user’s health. Currently, many smartwatches or bracelets monitor vital signs. Still, most of them are not developed to comply with medical device regulations, so the data they provide cannot be considered reliable. The difficulty involved in the process of validation and certification of medical devices means that manufacturers do not certify them as such and instead add disclaimers on their products indicating that they are not diagnostic devices. This development aims to provide a wristband that offers reliable diagnosis and accurate results. This is achievable as the proposed methodology allows us to address in a clear and structured way the technical and regulatory requirements that this type of device has to face. In terms of its category, it is a device classified by MDR as IIb and IEC 62304 as class B.

The system uses Bluetooth communication to monitor vital signs and other metrics remotely. In this way, it provides healthcare specialists with enough information to prevent and detect possible illnesses or accidents in patients who are not in a controlled environment, such as a hospital. The device incorporates cutting-edge, low-cost and low-power technology. It is designed to be user-friendly, and a fully automated measurement system requiring minimal user interaction.

Phases 1 and 2 of the proposed methodology have been fully implemented in this development. However, phase 3 is still in progress. The development is in the validation phase, where usability tests have been carried out in a real environment, namely in a nursing home for the elderly. Clinical trials and the certification process of the device are still pending.

#### 5.1.1. Methodology Phase 1—Development Feasibility

The implementation of the methodology starts with the execution of phase 1. Firstly, the existing technologies and solutions for these types of devices are reviewed, and the critical elements are identified: the measurement subsystem, in particular the measurement of oxygen, heart rate and the performance of electrocardiograms. After this, the first approximation of the solution is performed to clear the uncertainties of the measurement subsystem.

The difficulty of measuring these signals means that the first phase focuses on validating the measurement feasibility of the Photoplethysmography (PPG) and Electrocardiography (ECG) processes. As a technical solution, optical PPG is proposed, as it makes it possible to measure both oxygen and heart rate. If the vessels are illuminated, the amount of reflected light is proportional to the oxygen and heart rate values. Using the ECG, recording the heart’s signals is possible, for which electrodes are necessary.

This system’s feasibility verification is carried out with a first electronics prototype, including the sensing stage. In addition, as the housing is particularly relevant for light emission and ECG performance, a 3D-printed version of the housing is used to validate the concept. Figure 16 shows the first prototype to validate the measurement principle.

The initial results obtained were not satisfactory. On the one hand, for the PPG, when the intensity of the emitted light was high, the housing could not isolate the LED channels from the photodiodes. Therefore, the need for spacers between the photodiodes was identified to isolate the light from the LEDs further. In addition, the received signal was too weak due to the housing thickness, which was too thick for the photodiodes to be in direct contact with the skin. For all these reasons, it was decided to use a Teflon sheet to simulate the isolation that should exist in the final housing between the different LED channels. A provisional PCB assembly was then made. After testing, it was concluded that using two LED/photodiode combinations was enough.

In the case of the ECG, although the design allowed obtaining the signal from the heart, it was too weak. Therefore, it was concluded that this was due to the area of the electrodes being too small. This hypothesis was validated by directly welding larger electrodes. The results obtained were satisfactory. Figure 17 shows the unfiltered PPG and ECG values. Once the project manager and product manager have validated the results, phase 2 of the development proceeds.

#### 5.1.2. Methodology Phase 2—Incremental and Iterative Prototyping

Following phase 2 of the proposed methodology, the functional development of the device is carried out incrementally and iteratively.

As a result of the incremental and iterative development of the functionality, a smart bracelet designed for the remote monitoring of the main vital signs in elderly or dependent patients in the home environment is obtained. This device is equipped with embedded electronics to perform photoplethysmography (calculation of heart rate and blood oxygen saturation), lead I electrocardiography (between left and right hand), and monitor body temperature and physical activity.

Measurements are automatically taken regularly with minimal patient intervention, making it as simple as possible. Patients place their right hand on the device when prompted by the wearable to perform the electrocardiogram. The collected data are stored in the internal memory and can be downloaded through USB using the charging platform or through Bluetooth communication. Table 2 lists the main technical characteristics of the wearable.

The electronics of the device are embedded in a square sphere. The electronics are divided into two connected PCBs; one is used for the sensing stage, and the other is used for the control stage. Both PCBs are held in the lower case, while the battery is in the upper case. A conventional, adjustable silicone strap is used to fasten it to the wrist. As shown in Figure 18, the LED indicator and the front electrodes are located on the front of the device. At the back of the device are the rear electrodes used for the photoplethysmography (LEDs and photodiodes). The charging and serial communication port is also located on the back of the wristband. Figure 18 shows the location of these elements.

#### 5.1.3. Methodology Phase 3—Medical Product Consolidation

Finally, the execution of phase 3 has begun, for which the V-model has been completed, consolidating all the development, verifying each component, and generating the required documentation.

The device has been tested to verify the reliability of the measurement results. Firstly, the temperature sensor’s accuracy has been determined by temperature tests in a climatic chamber. Regarding the SpO2 measurement, a signal-to-noise ratio measurement was performed by placing the device on the charging station. This dock is designed to block external light artefacts and reflect the signal from the LEDs. In addition, both SpO2 and ECG measurements have been verified using a calibrated measuring device. Angular speed and linear acceleration were tested by driving the device on the axis of a controlled motor. Finally, the inclination has been checked against a calibrated measuring device.

The development is currently in the validation phase. On the one hand, usability tests have been carried out on dependent and older people living in an assisted living facility. On the other hand, environmental, electrical safety, and clinical trials, among others, are still pending. Those tests that must be performed in official institutions are still pending.

### 5.2. Use Case 2: Medical Device for Professional Use

This use case involves the use of the methodology in developing a medical device for professional use. The client and owner of the development is a start-up company that does not have enough technical expertise to develop the product entirely. Therefore, they decided to outsource part of it. The client’s contribution to the development is strictly limited to the definition of the algorithm that is the basis of the detection performed by the device.

Specifically, it is an embedded device, including embedded software and hardware. It is intended to be sold in Europe and is classified by the MDR as Class IIa. The software can be classified as class B, according to IEC 62304.

During this development, phases 2 and 3 of the methodology have been applied. Phase 1 was not executed because the client already had a preliminary prototype with which the feasibility of the development was validated.

The application of the proposed methodology can be considered a success, as this product is currently in the process of CE marking and has already successfully passed all the validation tests, including electrical safety tests, environmental tests and clinical trials on patients. The design is expected to be validated, and CE is marked by the competent authorities in the coming months.

### 5.3. Use Case 3: In Vitro Medical Device for Professional Use

In this use case, the methodology has been applied in its entirety. During the first phase, the main feasibility concerns and the selection of critical components were resolved. In the second phase, the incremental and iterative development of all the functionalities associated with the device was carried out. Finally, during the third phase, the product was consolidated through an intense documentation and verification process.

The company that owns the product is well established and has been commercialising products in Europe for several years. However, it does not have the resources to design and develop the device, so it decides to outsource this activity. This device has been designed and developed for professional use in the European market. It is classified as class B by the IVDR and class C by IEC 62304.

The development has required the design and development of embedded software and hardware. Currently, the product is CE marked and sold in Europe for some months. During the environmental, safety and clinical testing phase, as well as the certification of the device, no problems were found with the work completed during the design and development phases. The client was also satisfied with the developed product and the management carried out during the design and development phases. Therefore, it can be concluded that this product has allowed the complete validation of the proposed methodology.

### 5.4. Use Case 4: In Vitro Medical Device for Home and Professional Use

In this case, the three phases of the methodology are applied. In contrast to the previous use case, it was not necessary to develop the entire device but rather to extend the functionality of an existing device.

The company, a consolidated entity in the healthcare market, does not have a technical department, so it decided to outsource this development. Specifically, the development includes the design and development of embedded software and hardware for a home and professional use device, which was classified by the IVDR as class B and IEC 62304 as class C.

This development is on the market in Europe and the United States, and no problems have been identified with applying the proposed methodology. Thanks to this use case, the methodology is validated not only for the integral design of new medical products. It is also used to evolve or extend the functionality of devices already on the market. Furthermore, even though this methodology aims to address European regulatory needs, following the proposed methodology, this device has been approved by the FDA.

### 5.5. Use Case 5: In Vitro Medical Device for Professional Use

The entire methodology has been applied in this development. This is a device for professional use, which is regulated by the European IVDR regulations as class B and IEC 62304 as class B. The client, a consolidated company, has a technical department with embedded software and hardware development capabilities, but due to a lack of available resources, it has decided to outsource this activity.

The development has been carried out following the proposed methodology. A first phase of feasibility verification of the measurement principle was carried out, which is followed by the development of the functionality and, finally, the consolidation process. During this last phase, the client actively participated in the validation of the device, assuming almost all of its execution as its own.

The device is now CE-marked and sold in Europe without any problems. The client is also delighted with the result obtained during the outsourcing process.

### 5.6. Other Use Cases

In addition to the use cases mentioned above, this methodology has also been partially applied in developing medical devices for several start-ups.

In many cases, the application of the methodology has been limited to the execution of phase 1, Development Feasibility, and phase 2, Incremental and Iterative Prototyping. In many cases, executing the product consolidation phase has not been possible. These start-ups still need to obtain enough funding and structure to undertake the product consolidation phase and its subsequent commercialisation. It is expected that in the coming months, several of these companies will continue with the development process and undertake the product consolidation phase to homologate and commercialise their devices.

### 5.7. Certification of the Methodology

In 2021, Tekniker obtained the ISO 13485 certification for the design and development of medical devices. For this, phase 3 of this methodology has been fully implemented throughout the design and development procedures of embedded electronic products carried out in the Electronics and Communications Unit.

The procedures referring to phase 3 of this methodology have been audited by the company SGS, thus certifying the suitability of this methodology for the design and development of embedded medical devices. In addition, both in January 2022 and 2023, the application of these procedures in medical device developments has been reviewed by SGS, and the outcome of the review has been successful.

## 6. Conclusions

The research work presented in this article provides a methodological approach to designing and developing embedded sensor systems for healthcare. This work highlights that the high number of applicable standards requires a methodology that eases compliance in an efficient and orderly way. It also shows that the current methods are not suitable for the sector’s needs nor do they help the nature of start-up companies.

For these reasons, a new methodology based on three stages has been introduced and validated. The aforementioned methodology has allowed start-ups and consolidated healthcare companies to develop their embedded medical devices successfully. In addition, the methodology presented helped the start-ups mentioned above minimise their economic investment during the early stages of the project, where technical uncertainty is high. Consequently, they succeeded in meeting their customer’s product requirements by providing a device that complies with European medical regulations. Finally, the validation of this methodology has been carried out through the presentation of use cases in which its application has been successful.

## Figures and Tables

**Figure 1 sensors-23-02578-f001:**
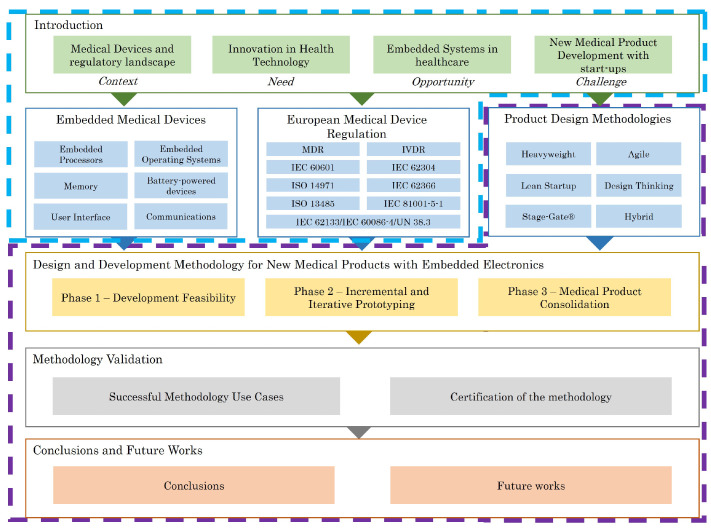
Article context. In blue are aspects addressed in the article [4], and in purple are those covered in this article.

**Figure 2 sensors-23-02578-f002:**
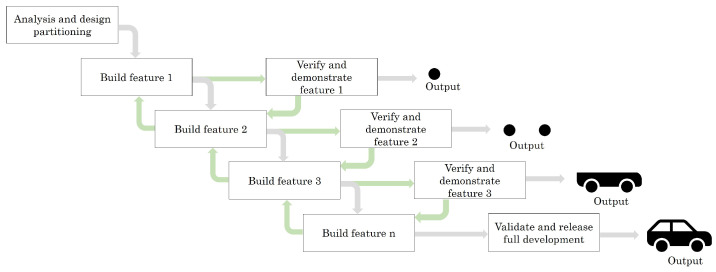
Incremental model and its outputs.

**Figure 3 sensors-23-02578-f003:**
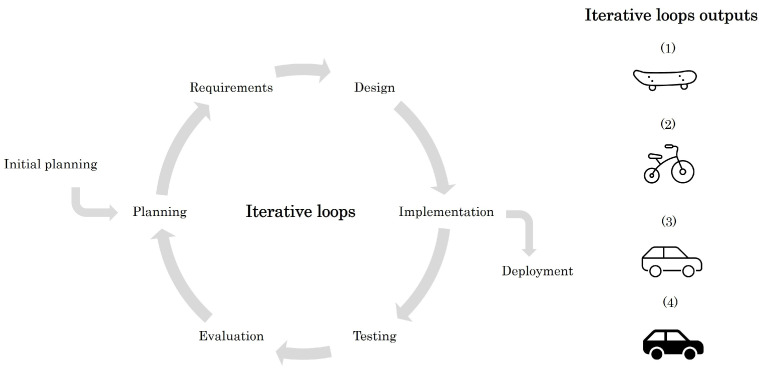
Iterative model and its four outputs identified as (1)–(4).

**Figure 4 sensors-23-02578-f004:**
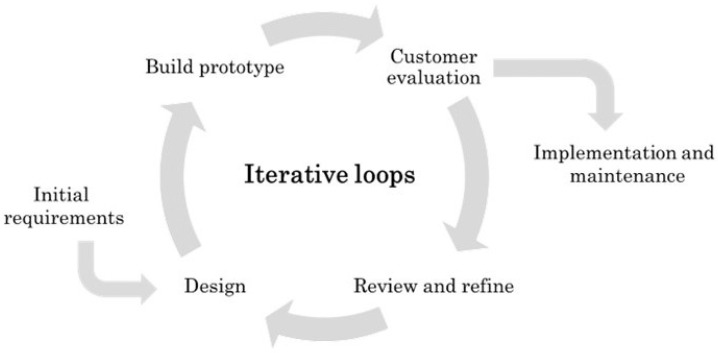
Prototype model.

**Figure 5 sensors-23-02578-f005:**
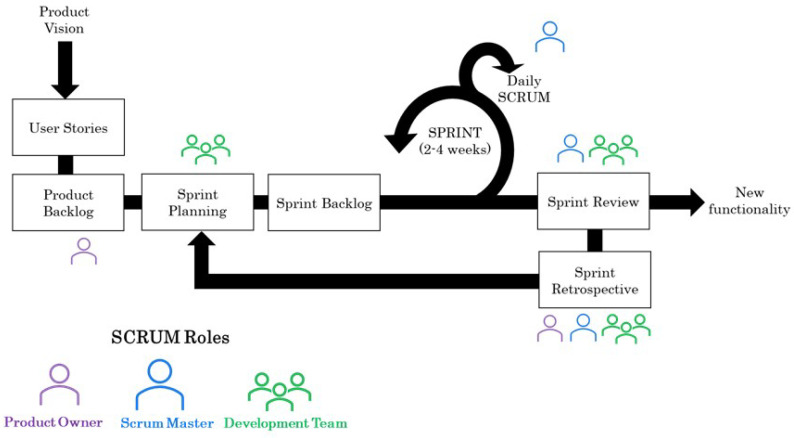
Scrum workflow.

**Figure 6 sensors-23-02578-f006:**
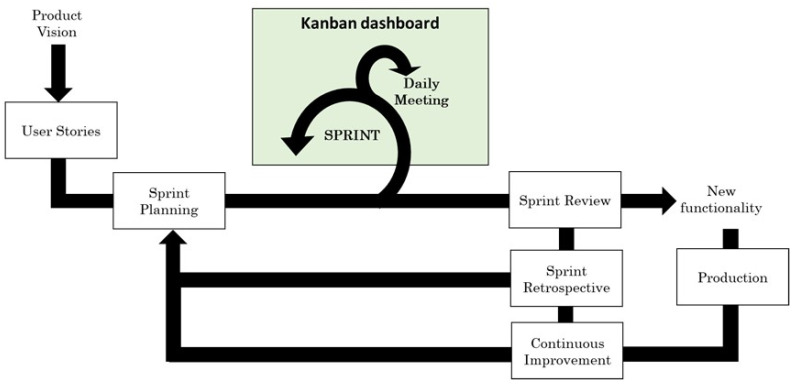
Scrumban workflow.

**Figure 7 sensors-23-02578-f007:**
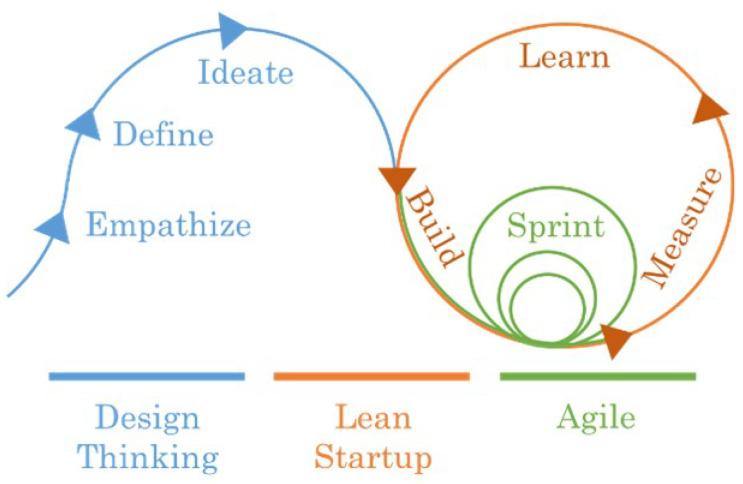
Generic Design Thinking–Lean Startup–Agile process.

**Figure 8 sensors-23-02578-f008:**
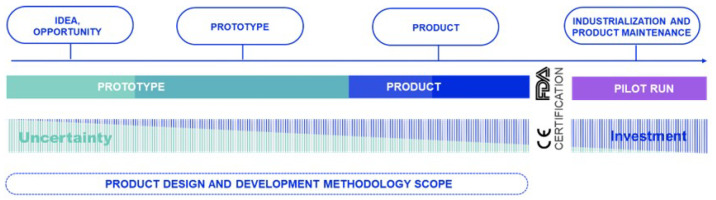
Scope of the proposed methodology.

**Figure 9 sensors-23-02578-f009:**
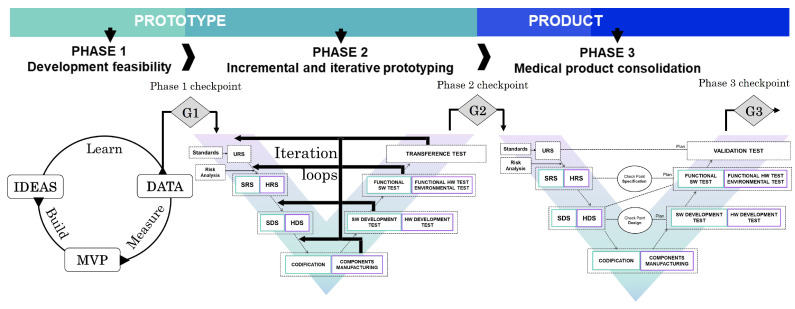
Stages of the proposed methodology.

**Figure 10 sensors-23-02578-f010:**
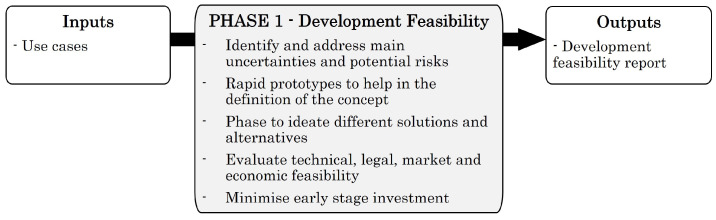
Development Feasibility phase: key characteristics, inputs and outputs.

**Figure 11 sensors-23-02578-f011:**
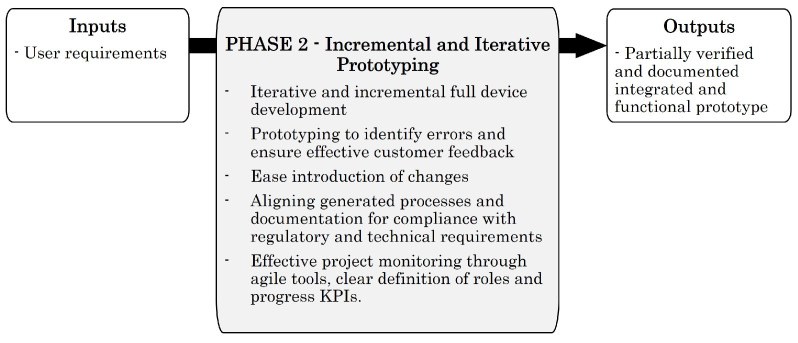
Incremental and Iterative prototyping phase: key characteristics, inputs and outputs.

**Figure 12 sensors-23-02578-f012:**
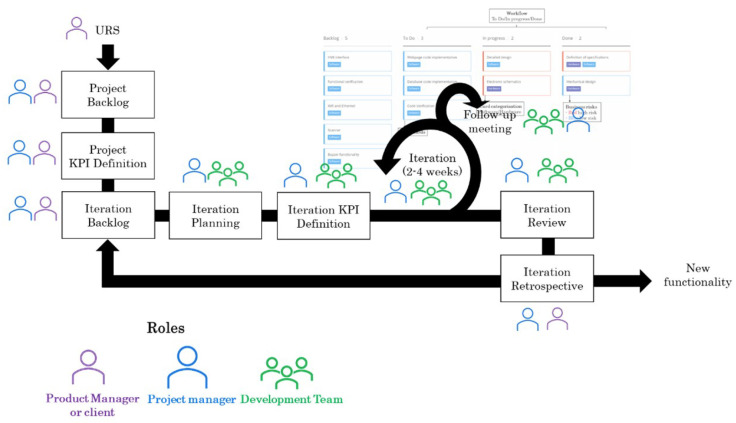
Proposed Agile methodology framework.

**Figure 13 sensors-23-02578-f013:**
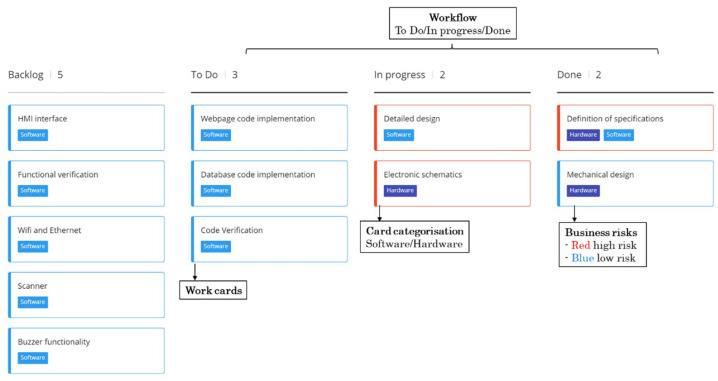
Proposed Agile methodology dashboard.

**Figure 14 sensors-23-02578-f014:**
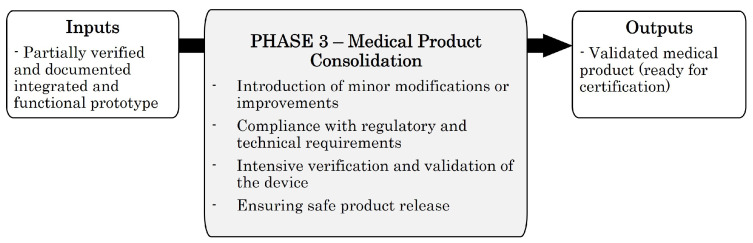
Medical Product Consolidation phase: key characteristics, inputs and outputs.

**Figure 15 sensors-23-02578-f015:**
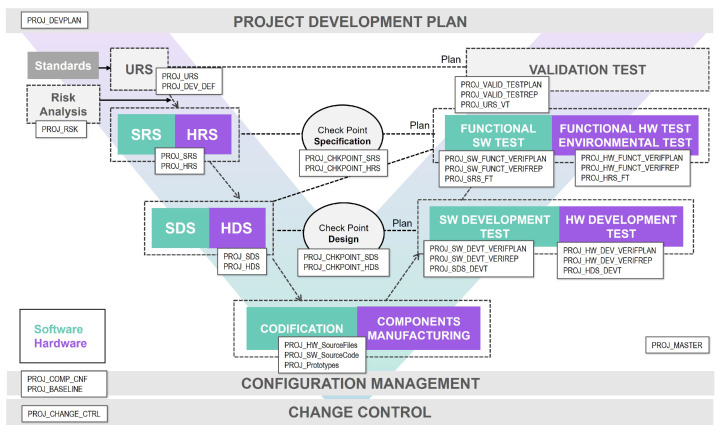
Deliverables associated to the phase 3.

**Figure 16 sensors-23-02578-f016:**
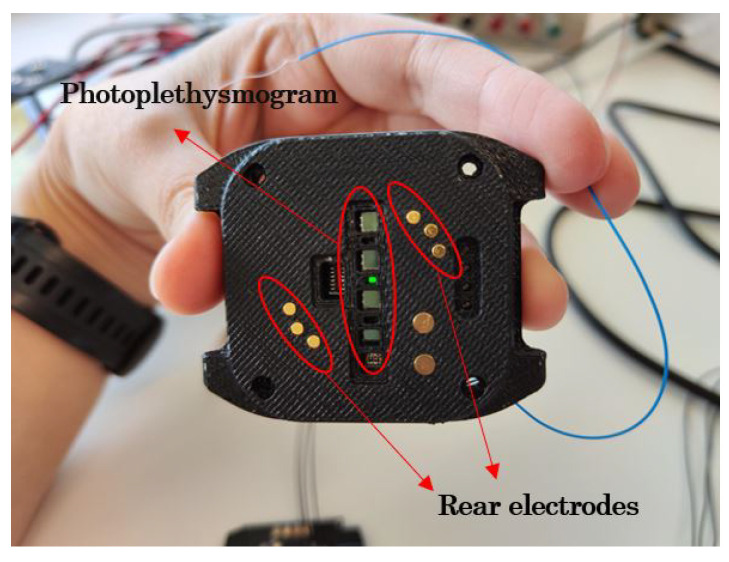
First prototype to validate feasibility.

**Figure 17 sensors-23-02578-f017:**
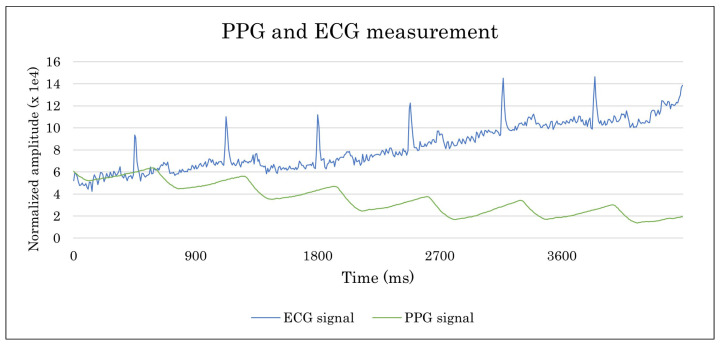
Measurements made with PPG and ECG sensor.

**Figure 18 sensors-23-02578-f018:**
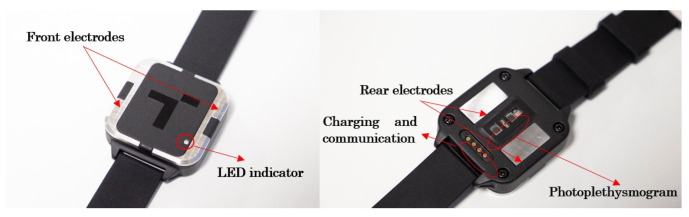
Front and rear side of the device.

**Table 1 sensors-23-02578-t001:** Contributions of each methodology to the proposed approach.

Contribution	Methodology
Divide the project into sub-tasks to allow the team to focus on a particular development and to better manage the development	Waterfall
Comply with the order of tasks and processes predefined in the medical device development standards	V-Model
Define the verification of each implementation phase prior to the deployment	V-Model
Enable incremental development to reduce technical and financial risk once the requirements are established.	Incremental
Detect errors in early stages of development	Incremental
Obtain client feedback in the early stages of development	Iterative
Develop iteratively during the initial phases of the project when requirements are not clearly defined	Iterative
Consider risk management in the different iterations of the methodology	Spiral
Develop rapid prototypes that help in the definition of the concept or clarifying technical uncertainties	Prototype
Parallelise tasks to minimise development time	Prototype
Ease the introduction of changes	Agile
Clearly define the acceptance criteria of each task or iteration	Agile
Review and reflection on the progress of iterations	Agile
Use a tool that allows the team to have a clear and complete overview of the status of the project at all times	Scrum
Define short iterations (Sprints) that deliver continuous value to the client	Scrum
Involve the team in the planning	Scrum
Define clear roles	Scrum
Identify the category, risk, and priority of each task	Kanban
Seek continuous improvement	Kanban
Do not impose drastic changes that may be difficult to accept	Kanban
Accept changes during development, at least partly	Kanban
Promote communication between the different members of the team	Crystal
Use technical tooling to assist the team during development and verification	Crystal
Measure development success and progress through KPIs	Lean Startup
Develop MVP to validate the concept	Lean Startup
Define a phase or process to ideate different solutions and alternatives	Design Thinking
Establish project check and evaluation points for the main phases of the project and its associated documentation	Stage-Gate

**Table 2 sensors-23-02578-t002:** Wearable’s technical specification.

Characteristic	Definition
User interface	RGB LED and Buzzer
Communication interface	USB and Bluetooth 5.1 (BLE)
Battery	370 mAh (1 week)
Internal flash memory	4 Gb
Size	49 mm × 45 mm × 15 mm
Electrocardiogram Interface	2 rear electrodes and 2 front electrodes
Photoplethysmography interface	1 Red LED, 1 Infrared LED, 1 green LED and 2 photodiodes

## Data Availability

Not applicable.

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
