# Peer review of "Medical Devices with Embedded Sensor Systems: Design and Development Methodology for Start-Ups"

_sensors, 2023, doi:10.3390/s23052578_

Round 1

Reviewer 1 Report

Paper is well written and conceptually also very well. Few typoes error are there but i hope authors are rewrite carefully. After these few changes the paper may get accept.

Author Response

Response to Reviewer 1 Comments

Point 1: Paper is well written and conceptually also very well. Few typoes error are there but i hope authors are rewrite carefully. After these few changes the paper may get accept.

Response 1: A thorough revision of the writing is carried out.

Author Response

Response to Reviewer 2 Comments

Point 1: Could the previous methodologies be employed for the development of such wearable devices sensor? What is the core novelty of this device? What are the advantages of the device over that other similar device developed by traditional methodologies?

Response 1: new concepts are added to clarify both the novelty and the difference with other devices on the market.

In the introduction:

“The design and development of embedded medical devices are regulated by two regulations, 2017/745, Medical Device Regulation (MDR) and 2017/746, In Vitro Medical Device Regulation (IVDR). These two regulations that have entered into force in 2022 regulate the design and development of these devices with more strict requirements than the previous directives. Devices already on the market will have to be recertified in accordance with these regulations, obliging healthcare companies to adapt their product design and development processes. Recognising the difficulty involved in adapting to the new requirements, on 6 January 2023, the European Commission has decided to extend the deadline by 3 to 4 years, depending on the risk of the device ~\ref{EuropeanCommision2023}.There are also other standards that define additional aspects such as functional safety (IEC 60601), software lifecycle (IEC 62304), quality management (ISO 13485), risk management (ISO 14971), usability (IEC 62366) and cybersecurity (IEC 81001-5-1), among others.

The large number of regulatory requirements demands a procedural methodology for performing such developments. After an analysis of existing product design methodologies, none has been identified that fully meets the needs of start-up companies wishing to develop medical devices. None of the identified methodologies is defined in a way that can address the three key characteristics of these developments: compliance with new European medical device regulations, development of innovative products with embedded hardware and software, and inexperienced companies in the sector.”

Wearable use case:

“The wearable solution involves a vital signs monitoring platform in the form of a smart wristband, which can collect relevant data about the user's health. Currently, there are a multitude of smart watches or bracelets that monitor vital signs, but most of them are not developed to comply with medical device regulations, so the data they provide cannot be considered reliable. The difficulty involved in the process of validation and certification of medical devices means that manufacturers do not certify them as such and instead add disclaimers on their products indicating that they are not diagnostic devices. The aim of this development is to provide a wristband that offers reliable diagnosis and accurate results. This is achievable as the proposed methodology allows to address in a clear and structured way the technical and regulatory requirements that this type of device has to face. In terms of its category, it is a device classified by MDR as IIb and by IEC 62304 as class B.”

Point 2: There are too many paragraphs that are very short. Merge some into longer and more logical paragraphs.

Response 2: done.

Point 3: Figures 1 and 2 is in low quality.

Response 3: exchanged for higher quality images.

Point 4: The first letters of many words in such subtitle as “3.6.1. Scrumban methodology”, “3.6. Hybrid methodologies”, “3.2.2. Kanban framework” should be capitalized. Check all that apply.

Response 4: all subtitles have been checked and corrected.

Point 5: There are two “use case 1”. 5.2. Medical device: use case 1 and 5.3. In-Vitro Medical Device: use case 1. Double check and clarify them.

Response 5: use case title strategy changed for better understanding.

Point 6: Professional proofreading is suggested and some typos needs to be taken care of.

Response 6: A thorough revision of the writing is carried out.

Reviewer 3 Report

This paper provided a methodology for the design and development of embedded medical devices aimed at minimizing the economic investment in the technology risk phase. By listing product design ideas and solutions in the context of current regulatory conditions. According to the actual needs of the market, methods based on the implementation of three stages is proposed which is quite interesting and meaningful. The flowcharts are presented in good forms.

Generally, the paper is well structured, and the research topic is also clearly addressed. I recommend to accept the manuscript.

Author Response

Response to Reviewer 3 Comments

Point 1: This paper provided a methodology for the design and development of embedded medical devices aimed at minimizing the economic investment in the technology risk phase. By listing product design ideas and solutions in the context of current regulatory conditions. According to the actual needs of the market, methods based on the implementation of three stages is proposed which is quite interesting and meaningful. The flowcharts are presented in good forms.

Generally, the paper is well structured, and the research topic is also clearly addressed. I recommend to accept the manuscript.

Response 1: A thorough revision of the writing is carried out.

Reviewer 4 Report

The authors presented a methodology to design and developed embedded medical devices, in compliance with the regulatory requirements applied to the development. In addition, they presented a successful use cases: wearable device for monitoring vital signs.

 #--- Detailed review ---#

1. Please improve the readability of the article. There are some ill constructed sentences that need to be fixed, for instance "The large number of standards to be met makes it necessary to have a development methodology that allows to carry out the work in an organised and controlled way.". Please do a full proofread reading

2. The abstract seems incomplete. Please put some results at the end. 1-2 sentences most.

3. I advise you to put the links in the below regulations:
    2017/745, Medical Device Regulation (MDR) and
    2017/746, In Vitro Medical Device Regulation (IVDR)
and, as they refer only to European regulation write "European Medical Devices Regulation: MDR 2017/745" .

4. Write out the meaning of the acronym first, such as SGS for instance

5. I don't think section 2 adds value to the article.

6. I don't understand the need for such an extensive review by the methodologies. A summary and, maybe a table with the main characteristics could be useful for the article. I remind you that this is not a survey.

7. The Figure 1 should have rights. In that sense, I recommend that you add the source. Besides, it is too small to make it difficult to read. What is mean (1) - (4) in Figure 3. This must be described in the text

8. Why all the phases of the Waterfall methodology are listed inline, and Spiral model not?

9. I think that if the authors' decision is to keep a detailed description of the methodologies they should maintain some consistency in the way they write. Some of them are inaccurate in their descriptions. Then, there are minor corrections to the form.

10. "IEC 60601 (Basic safety and essential performance), 609 IEC 62304 (Medical device software), ISO 14971 (Medical device risk management), IEC 610 62366 (Medical device usability), ISO 13485 (Medical device quality management systems), 611 IEC 81001-5-1 (Medical device cybersecurity) " this is mentioned before twice, I think.

11. "proof of concept," and after the authors use "Development feasibility". You must be consistent with the terms. Also, and since you enumerated the methodology, my suggestion is to use enumerate instead of items.

12. I would eliminate the background colors in Figure 9, because it becomes difficult to read. Plus, what do G1, G2, and G3 mean? It would help to have a legend for the acronyms.
I realized that Figure 15 is included in 9. However, in general, the figures are barely understandable.

13. I don't see the point of Figures 10, 11, 14. Could you explain?

14. Chapter 4 is too descriptive, making it hard to follow. Maybe a diagram illustrating the steps. In general, the paper is too long.

15. "Tekniker, a Research and Development Centre, ". It has already been said before, it is redundant. "(...)  ISO 13485 certification process." also.

16. Have you performed any tests with a certified wearable? Haven't you performed any user tests? Is it a single measure, or an average? How can you validate this result?

17. Figure 17: all plots needs units. Another sugestion is to remove all zeros, instead of 20000 use 2e4

The work focused on the implementation of a methodology for the design and development of an embedded sensor system for healthcare. At this point, the document seems too descriptive, making it difficult to understand. This is followed by the presentation of 3 use cases. Here, I could not make the connection with the previous point. And, I could not find any differences or similarities between them in a figure or table. Are they isolated cases?  FDA approved equipment has been mentioned, but we do not know which one or which measurements it obtained.

Some typos:

wearable monitoring devices...   -> wearable monitoring devices. (avoid "dots")
presented in figure 1. -> presented in Figure 1.
That is because only -> That is, because only
a paper titled -> an article titled (be consistent!)
these are tests are related -> these tests are related
Normally the main use -> Normally, the main use
Bluetooth communication -> bluetooth communication
hear rate <- heart rate

REGULATION (EU) 2017/745 OF THE EUROPEAN PARLIAMENT AND OF THE COUNCIL capslock?!?!

.... There are others later in the text

error:

https://doi.org/10.1007/978-3-319-69926-4_19/FIGURES/3
https://doi.org/10.1108/17538371211269031/FULL/XML

https://doi.org/10.14257/IJSEIA.2015.9.11.05 redirects to here:

https://doi.org/10.1007/978-3-319-76998-1_14/COVER

https://doi.org/10.37591/JoMEA redirects to here:
https://engineeringjournals.stmjournals.in/index.php/JoMEA/index

.... I have not seen all!

Author Response

Response to Reviewer 4 Comments

Point 1: Please improve the readability of the article. There are some ill constructed sentences that need to be fixed, for instance "The large number of standards to be met makes it necessary to have a development methodology that allows to carry out the work in an organised and controlled way.". Please do a full proofread reading.

Response 1: done.

Point 2: The abstract seems incomplete. Please put some results at the end. 1-2 sentences most.

Response 2: done.

Point 3: I advise you to put the links in the below regulations:

    2017/745, Medical Device Regulation (MDR) and

    2017/746, In Vitro Medical Device Regulation (IVDR)

and, as they refer only to European regulation write "European Medical Devices Regulation: MDR 2017/745" .

Response 3: done “The design and development of embedded medical devices are regulated by two regulations, European Medical Devices Regulation: 2017/745 MDR [1] and 2017/746 IVDR [2].”

Point 4: Write out the meaning of the acronym first, such as SGS for instance

Response 4: done.

Point 5: I don't think section 2 adds value to the article.

Response 5: The regulatory context is centralised in section 2. To this end, other references to standards throughout the text are removed.

Point 6: I don't understand the need for such an extensive review by the methodologies. A summary and, maybe a table with the main characteristics could be useful for the article. I remind you that this is not a survey.

Response 6: the writing of the methodologies of the state of the art is unified, focusing on their contribution to the new methodology. A table with the key contributions is also added.

Point 7: The Figure 1 should have rights. In that sense, I recommend that you add the source. Besides, it is too small to make it difficult to read. What is mean (1) - (4) in Figure 3. This must be described in the text

Response 7: some images are expanded and a reference to figure 1 is added. The meaning of (1)-(4) in the title of figure 3 is added: “Iterative model and its 4 outputs identified as (1)-(4)”

Point 8: Why all the phases of the Waterfall methodology are listed inline, and Spiral model not?

Response 8: style is unified.

Point 9: I think that if the authors' decision is to keep a detailed description of the methodologies they should maintain some consistency in the way they write. Some of them are inaccurate in their descriptions. Then, there are minor corrections to the form.

Response 9: the writing of the methodologies of the state of the art is unified, focusing on their contribution to the new methodology. A table with the key contributions is also added.

Point 10: "IEC 60601 (Basic safety and essential performance), 609 IEC 62304 (Medical device software), ISO 14971 (Medical device risk management), IEC 610 62366 (Medical device usability), ISO 13485 (Medical device quality management systems), 611 IEC 81001-5-1 (Medical device cybersecurity) " this is mentioned before twice, I think.

Response 10: The regulatory context is centralised in section 2. To this end, other references to standards throughout the text are removed.

Point 11: "proof of concept," and after the authors use "Development feasibility". You must be consistent with the terms. Also, and since you enumerated the methodology, my suggestion is to use enumerate instead of items.

Response 11: the consistency of the terms is reviewed, and the enumeration is used to list the different phases.

Point 12: . I would eliminate the background colors in Figure 9, because it becomes difficult to read. Plus, what do G1, G2, and G3 mean? It would help to have a legend for the acronyms.
I realized that Figure 15 is included in 9. However, in general, the figures are barely understandable.

Response 12: The meaning of G1, G2 and G3 is clarified. Background colours are removed and Figure 15 is simplified to avoid redundancies with Figure 9.

Point 13: I don't see the point of Figures 10, 11, 14. Could you explain?

Response 13: Figures 10, 11 and 14 are edited to show the key points of each phase.

Point 14: Chapter 4 is too descriptive, making it hard to follow. Maybe a diagram illustrating the steps. In general, the paper is too long.

Response 14: is rewritten to make it more readable. Figures 10, 11 and 14 are edited to show the key points of each phase.

Point 15: "Tekniker, a Research and Development Centre, ". It has already been said before, it is redundant. "(...)  ISO 13485 certification process." also.

Response 15: duplications in the text are eliminated.

Point 16: Have you performed any tests with a certified wearable? Haven't you performed any user tests? Is it a single measure, or an average? How can you validate this result?

Response 16: detailed information on how the device has been verified is added.

“The device has been tested to verify the reliability of the measurement results. Firstly, the accuracy of the temperature sensor has been determined by performing temperature tests in a climatic chamber. Regarding the SpO2 measurement, a signal to noise ratio measurement was performed by placing the device on the charging station. This dock is designed to block external light artifacts and reflect the signal from the LEDs. Also, both SpO2 and ECG measurements have been verified using a calibrated measuring device. Angular speed and linear acceleration were tested by driving the device on the axis of a controlled motor. Finally, the inclination has been checked against a calibrated measuring device.”

Point 17: Figure 17: all plots needs units. Another sugestion is to remove all zeros, instead of 20000 use 2e4

Response 17: done.

Point 18: The work focused on the implementation of a methodology for the design and development of an embedded sensor system for healthcare. At this point, the document seems too descriptive, making it difficult to understand. This is followed by the presentation of 3 use cases. Here, I could not make the connection with the previous point. And, I could not find any differences or similarities between them in a figure or table. Are they isolated cases?  FDA approved equipment has been mentioned, but we do not know which one or which measurements it obtained.

Response 18: titles and text have been restructured to make the validation stage of the methodology clearer. In addition, a figure is added specifying article context and the contribution of each section.

Point 19: Some typos

Response 19: done.

Round 2

Reviewer 2 Report

accept

Author Response

The text has been reviewed to correct spelling and syntactical errors.

Reviewer 4 Report

The authors presented a methodology to design and developed embedded medical devices, in compliance with the regulatory requirements applied to the development. In addition, they presented a successful use cases: wearable device for monitoring vital signs.

 #--- Detailed review ---#

1. The authors made the requested corrections, and answered correctly all the questions asked.

2. The sentence "Development feasability, ... " is redundant, because is listed after.

3. I advise you to carefully re-read the entire document to correct these minor syntax errors (read the list below).

Typos:
Programmable Electrical Medical Medical -> Programmable Electrical Medical
BarryW. Boehm in 1986 -> Barry W. Boehm in 1986
agile methodologies dates -> Agile methodologies dates
monitored or or controlled -> monitored or controlled
iteration.If some -> iteration. If some
deliverables, Regarding phase 3 -> deliverables, regarding phase 3
the photodiodes. (...) the photo-diodes -> the photodiodes. (...) the photodiodes

Author Response

(The authors gave the same response as above.)
